# Fortify the Guardian, Not the Treasure: Resilient Adversarial Detectors

## Abstract

This paper presents `RADAR`—**R**obust **A**dversarial **D**etection via **A**dversarial **R**etraining—an approach designed to enhance the robustness of adversarial detectors against *adaptive attacks*, while maintaining classifier performance. An adaptive attack is one where the attacker is aware of the defenses and adapts their strategy accordingly. Our proposed method leverages adversarial training to reinforce the ability to detect attacks, without compromising clean accuracy. During the training phase, we integrate into the dataset adversarial examples, which were optimized to fool *both* the classifier *and* the adversarial detector, enabling the adversarial detector to learn and adapt to potential attack scenarios. Experimental evaluations on CIFAR-10, SVHN and ImageNet datasets demonstrate that our proposed algorithm significantly improves a detector's ability to accurately identify adaptive adversarial attacks—without sacrificing clean accuracy.

## 1 Introduction

Rapid advances in deep learning systems and their applications in various fields such as finance, healthcare, and transportation, have greatly enhanced human capabilities (Huang et al., 2020; Miotto et al., 2018; Chen et al., 2024a; Shamshirband et al., 2021; Lv et al., 2020). However, these systems continue to be vulnerable to adversarial attacks, where unintentional or intentionally designed inputs—known as *adversarial examples*—can mislead the decision-making process (Goodfellow et al., 2014; Szegedy et al., 2013; Tamam et al., 2023; Chen et al., 2024b; Lapid et al., 2024; Carlini & Wagner, 2017b; Lapid & Sipper, 2023b; Andriushchenko et al., 2020; Lapid et al., 2022; Lapid & Sipper, 2023a). Such adversarial attacks pose severe threats both to the usage and trustworthiness of deep learning technologies in critical applications (Finlayson et al., 2019; Liu et al., 2023).

Extensive research efforts have been dedicated to developing robust machine learning classifiers that are resilient to adversarial attacks. One popular and effective way relies on *adversarial training*, which involves integrating adversarial examples within the training process, to improve the classifier's resistance to attack (Madry et al., 2017; Zhao et al., 2022). However, the emergence of adversarial detectors designed to identify malicious inputs (Lu et al., 2017; Grosse et al., 2017a; Gong & Wang, 2023; Lust & Condurache, 2020; Metzen et al., 2016) has opened new avenues for enhancing or thwarting the security of AI systems.

This paper presents a novel approach to adversarially train detectors, combining the strengths both of classifier robustness and detector sensitivity to adversarial examples.

*Adversarial detectors* (Figure 1), which aim to distinguish adversarial examples from benign ones, have gained momentum recently, but their robustness is unclear. The purpose of adversarial detection is to enhance the robustness of machine learning systems by identifying and mitigating adversarial attacks—thereby preserving the reliability and integrity of the classifiers in critical applications.

Several studies have found that these detectors—even when combined with robust classifiers that have undergone adversarial training—can be fooled by designing tailored attacks; this engenders a cat-and-mouse cycle of attacking and defending (Carlini & Wagner, 2017a; Grosse et al., 2017b). It is therefore critical to

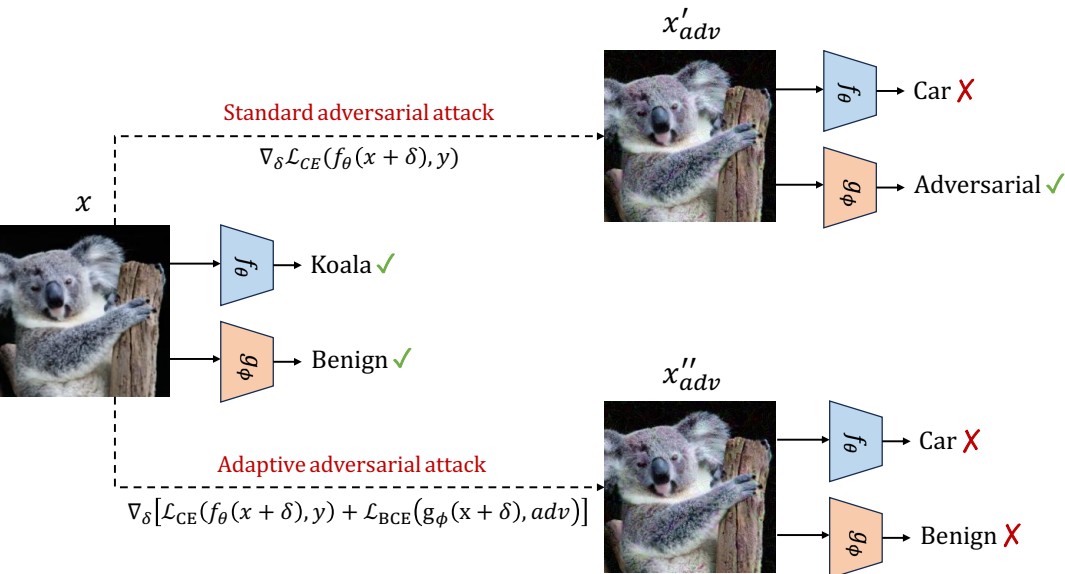

Figure 1: General scheme of adversarial attacks. $x$: original image. $x'_{\text{adv}}$: standard adversarial attack. $x''_{\text{adv}}$: adaptive adversarial attack, targeting both $f_\theta$ (classifier) and $g_\phi$ (detector). The attacker's goal is to fool the classifier into misclassifying the image *and* at the same time fool the detector into reporting the attack as benign (i.e., fail to recognize an attack).

thoroughly assess the robustness of adversarial detectors and ensure that they are able to withstand adaptive adversarial attacks.

**The attacker's goal is to fool the classifier into misclassifying the image *and* at the same time fool the the detector into reporting the attack as benign (i.e., fail to recognize an attack).** The attacker, in essence, carries out a two-pronged attack, targeting both the "treasure" and the treasure's "guardian".

An *adaptive* adversarial attack is *aware of the defenses*—and attempts to bypass them. To our knowledge, research addressing such attacks is lacking.

We thus face a crucial problem: How can adversarial detectors be made robust? This question underscores the need for a comprehensive understanding of the interplay between adversarial-training strategies and the unique characteristics of adversarial detectors.

We hypothesize that adversarially training adversarial detectors will lead to their improvement both in terms of robustness and accuracy, thus creating a more-resilient defense mechanism against sophisticated adversarial attacks. This supposition forms the basis of our investigation and serves as a guiding principle for the experiments and analyses presented in this paper.

There are two major motivations to our work herein:

1. Contemporary adversarial detectors are predominantly evaluated within constrained threat models, assuming attackers lack information about the adversarial detector itself. However, this approach likely does not align with real-world scenarios, where sophisticated adversaries possess partial knowledge of the defense mechanisms.

We believe it is necessary to introduce a more-realistic evaluation paradigm that considers the potential information available to an attacker. Such a paradigm will improve—perhaps significantly—the reliability of adversarial-detection mechanisms.

2. Our fundamental assumption is that adversarial training of the *adversarial detector*—in isolation—has the potential to render the resultant system significantly more challenging for adversaries to thwart. Unlike conventional approaches, where increasing the robustness of the *classifier* often leads to decreased accuracy, our approach focuses solely on enhancing the capabilities of the detector, thereby avoiding such trade-offs. By bolstering the detector through adversarial training, we introduce a distinct tier of defense that adversaries must contend with, amplifying the overall adversarial resilience of the system.

This paper focuses on the adversarial training of a defensive adversarial detector to improve its resilience against robust and sophisticated adversarial attacks. **We suggest a paradigm shift: instead of defending the classifier—strengthen the adversarial detector**. Thus, we ask:

> **(Q)** *How can adversarial detectors be made robust?*

To our knowledge, problem **(Q)** remains open in the literature. Our contributions are:

- Paradigm shift: Unlike existing approaches that focus on robustifying classifiers, we introduce a novel paradigm of robustifying adversarial detectors, eliminating the trade-off between clean and robust classification.

- We introduce `RADAR`, a pioneering adversarial training technique specifically designed to enhance the resilience of adversarial detectors, demonstrating its effectiveness through extensive evaluations on three datasets and across six detection architectures.

- Our study provides comprehensive insights into the generalizability and efficacy of adversarial training strategies across diverse data distributions and model architectures, reinforcing the critical role of robust adversarial detectors in securing machine learning systems.

The next section describes previous work on adversarial training. Section 3 delineates our methodology, followed by the experimental framework in Section 4. We describe our results in Section 5, ending with conclusions in Section 6.

## 2 Previous Work

The foundation of adversarial training was laid by Madry et al. (2017), who demonstrated its effectiveness on the MNIST (LeCun et al., 1998) and CIFAR-10 (Krizhevsky, 2009) datasets. Their approach used Projected Gradient Descent (PGD) attacks to generate adversarial examples and incorporate them into the training process alongside clean (unattacked) data.

Subsequent work by Kurakin et al. (2016) explored diverse adversarial training methods, highlighting the trade-off between robustness and accuracy. These early pioneering works laid the groundwork for the field of adversarial training and showcased its potential for improving classifier robustness.

Adversarial training has evolved significantly since its inception. Zhang et al. (2017) introduced MixUp training, probabilistically blending clean and adversarial samples during training, achieving robustness without significant accuracy loss.

Tramer & Boneh (2019) investigated multi-attack adversarial training, aiming for broader robustness against diverse attack types.

A number of works explored formal-verification frameworks to mathematically certify classifier robustness against specific attacks, demonstrating provably robust defense mechanisms (Cohen et al., 2019; Lecuyer et al., 2019; Li et al., 2023; Singh et al., 2018).

Zhang et al. (2019) provided theoretical insights into the robustness of adversarially trained models, offering a deeper understanding of the underlying mechanisms and limitations. Their work elucidated the trade-offs between robustness, expressiveness, and generalization in adversarial training paradigms, contributing to a more comprehensive theoretical framework for analyzing and designing robust classifiers.

In the pursuit of improving robustness against adversarial examples, Xie et al. (2019) proposed a novel approach using feature denoising. Their work identified noise within the extracted features as a vulnerability during adversarial training. To address this, they incorporated denoising blocks within the network architecture of a convolutional neural network (CNN). These blocks leverage techniques such as non-local means filters to remove the adversarial noise, leading to cleaner and more-robust features. Their approach achieved significant improvements in adversarial robustness on benchmark datasets, when compared to baseline adversarially trained models. This work highlights feature denoising as a promising defense mechanism that complements existing adversarial training techniques. Pinhasov et al. (2024) introduced a novel method to combat adversarial attacks on deepfake detectors. The authors leveraged eXplainable Artificial Intelligence (XAI) (Došilović et al., 2018) to generate interpretability maps, revealing the decision-making process of the deepfake detector. By analyzing these maps, their system could identify vulnerabilities exploited by adversarial attacks.

Building on the limitations of standard adversarial training, requiring a large amount of labeled data, Carmon et al. (2019) explored leveraging unlabeled data for improved robustness. Their work demonstrates that incorporating unlabeled data through semi-supervised learning techniques, such as self-training, can significantly enhance adversarial robustness. This approach bridges the gap in sample complexity, achieving high robust accuracy with the same amount of labeled data needed for standard accuracy. Their findings were validated empirically on datasets such as CIFAR-10, achieving state-of-the-art robustness against various adversarial attacks. This research highlights the potential of semi-supervised learning as a cost-effective strategy to improve adversarial training and achieve robust models.

Despite its successes, adversarial training faces several challenges. Transferability of adversarial examples remains a concern (Cheng et al., 2019; Dong et al., 2021), because attacks often lack effectiveness across different classifiers and architectures. The computational cost of generating and training on adversarial examples can be significant, especially for large models and datasets. Moreover, adversarial training can lead to a reduction in clean-data accuracy, requiring effective optimization strategies to mitigate this trade-off (Yang et al., 2020). Additionally, carefully crafted evasive attacks can sometimes bypass adversarially trained classifiers, highlighting the need for further research and diversification of defense mechanisms.

There are but a few examples of works that compare adversarial attacks that take into account full knowledge of the defense, i.e., adaptive attacks. He et al. (2022) showed promising results, at the cost of multiple model inferences coupled with multiple augmented neighborhood images. Klingner et al. (2022) demonstrated a detection method based on edge extraction of features such as depth estimation and semantic segmentation, and comparison to natural image edges based on the SSIM metric. We believe the underlying assumption of adversarial examples containing abnormal edges might not hold in unnatural settings.

Previous research has explored using sentiment analysis for adversarial example detection (Wang et al., 2023), focusing on the impact of adversarial perturbations on deep neural networks' hidden-layer feature maps under attack.

All of the above compared their work with some adaptive power like PGD, yet didn't show state-of-the-art adaptive attacks, which optimize under the constraint imposed by the classifier decision boundary, like SPGD and OPGD (Bryniarski et al., 2021). Only a few methods were compared under these powerful optimizers. Yang et al. (2022a) presented a novel encoder-generator architecture that compared the generated image label with the original image label. We believe this was a step in the right direction, yet with a computationally expensive generator that had difficulties differentiating semantically close classes. We compared our results to theirs, and others (Shan et al., 2020; Sperl et al., 2020; Tian et al., 2021), which already have some comparison to SPGD or OPGD.

# 3 Methodology

Before delineating the methodology we note that the overall framework comprises three distinct components:

1. **Classifier**: Which is not modified (i.e., no learning).

2. **Detector**: With a "standard" loss function (to be discussed).

3. **Attacker**: With an adaptive loss function (to be discussed).

## 3.1 Threat Model

We assume a strong adversary, with full access both to the classifier $f$ and the adversarial detector $g$. The attacker possesses comprehensive knowledge of the internal workings, parameters, and architecture of both models.

The attacker has two goals: **1)** To manipulate the classifier's ($f$) class prediction $y$ such that it outputs a class different from the correct class, i.e., $f(x) \neq y$. Additionally, the adversarial noise $\delta$ should be sufficiently small, constrained through an upper bound $\epsilon$ on the $l_p$ norm. This problem can be formulated as:

$$\underset{k=0,1,\dots K-1}{\arg\max} \ f_k(x + \delta) \neq y, \ (x, y) \in \mathcal{D}, \ \text{s.t.} \ \|\delta\|_p \leq \epsilon, \tag{1}$$

where $K$ denotes the number of classes in the classification task. We can solve this problem using the following optimization objective:

$$\underset{\delta: \|\delta\|_p \leq \epsilon}{\arg\max} \ \mathcal{L}_{\text{CE}}(f_\theta(x + \delta), y), \tag{2}$$

where $\mathcal{L}_{\text{CE}}$ is the cross-entropy loss.

The attacker's second goal is: **2)** To cause the detector $g$ to predict that the image is benign, i.e., $g(x) \neq \texttt{adv}$, while also maintaining the $l_p$ norm constraint:

$$\underset{k=\texttt{ben},\texttt{adv}}{\arg\max} \ g_k(x + \delta) \neq \texttt{adv}, \ x \in \mathcal{D}, \ \text{s.t.} \ \|\delta\|_p \leq \epsilon. \tag{3}$$

We can solve this problem using the following optimization objective:

$$\underset{\delta: \|\delta\|_p \leq \epsilon}{\arg\max} \ \mathcal{L}_{\text{BCE}}(g_\phi(x + \delta), \texttt{adv}), \tag{4}$$

where $\mathcal{L}_{\text{BCE}}$ is the binary cross-entropy loss and $\epsilon$ is the allowed perturbation norm.

Combining the two goals, the attacker's overall goal is defined as:

$$\underset{\delta: \|\delta\|_p \leq \epsilon}{\arg\max} \ \mathcal{L}_{\text{CE}}(f_\theta(x + \delta), y) + \mathcal{L}_{\text{BCE}}(g_\phi(x + \delta), \texttt{adv}). \tag{5}$$

## 3.2 Problem Definition

We now formalize the problem of enhancing the robustness of an adversarial detector ($g$) through adversarial training. The setup involves a classifier ($f$) and an adversarial detector ($g$), both initially trained on a clean dataset $\mathcal{D}$ containing pairs $(x, y)$, where $x$ is an input data point and $y$ is the ground-truth label. We denote by $\theta$ and $\phi$ the weight parameters of the classifier and the detector, respectively. Note that $\mathcal{D}$ is the clean dataset—adversarial examples are represented by $x + \delta$.

Adversarial training is formulated as a minimax game:

$$\min_\theta \left\{ \mathbb{E}_{(x,y) \sim \mathcal{D}} \left[ \max_{\delta: \|\delta\|_p \leq \epsilon} \mathcal{L}_{\text{CE}}(f_\theta(x + \delta), y) \right] \right\}. \tag{6}$$

This minimax game pits the classifier against an adversary: the adversary maximizes the classifier's loss on perturbed inputs, while the classifier minimizes that loss, across all such perturbations, ultimately becoming robust to attacks.

Our main objective is to improve the robustness of detector $g$ using adversarial training, by iteratively updating $\phi$ based on adversarial examples. Formally, the optimization objective for the adversarial training of the detector is:

$$\min_{\phi} \left\{ \mathbb{E}_{(x,y)\sim\mathcal{D}} \left[ \max_{\delta:\|\delta\|_p\leq\epsilon} \mathcal{L}_{\mathrm{BCE}}(g_{\phi}(x+\delta), \mathtt{adv}) \right] \right\}. \tag{7}$$

As with the classifier, we have a minimax game for the detector.

However, the above equation does not take into account the classifier, $f$. So we added the classifier outputs to our (almost) final objective:

$$\min_{\phi} \left\{ \mathbb{E}_{(x,y)\sim\mathcal{D}} \left[ \max_{\delta:\|\delta\|_p\leq\epsilon} (\mathcal{L}_{\mathrm{CE}}(f_{\theta}(x+\delta), y) + \mathcal{L}_{\mathrm{BCE}}(g_{\phi}(x+\delta), \mathtt{adv})) \right] \right\}. \tag{8}$$

Here, the objective incorporates both the classifier and the detector in a unified adversarial-training framework. The goal is to simultaneously minimize the classification error *and* enhance the detector's robustness.

However, a potential conflict arises with this combined objective. The cross-entropy loss ($\mathcal{L}_{\mathrm{CE}}$) for the classifier and the binary cross-entropy loss ($\mathcal{L}_{\mathrm{BCE}}$) for the detector inherently contradict each other. The attacker's objective is to simultaneously induce misclassification in the classifier and deceive the detector into labeling adversarial examples as benign. This dual objective creates opposing forces in the optimization process. Specifically, the attacker's optimization of $\mathcal{L}_{\mathrm{CE}}$ aims to increase the classification error by adjusting $\delta$ so that the predicted class probabilities do not align with the true labels. Conversely, the optimization of $\mathcal{L}_{\mathrm{BCE}}$ seeks to reduce the detector's ability to differentiate between clean and adversarial examples, driving $g_{\phi}$ to misclassify adversarial inputs as benign. This opposition can lead to conflicting gradient directions, where optimizing for one objective may detract from progress on the other, thereby complicating the optimization process and potentially preventing either objective from being fully achieved.

This dual optimization can result in non-convergent behavior for several reasons. Firstly, the gradients derived from $\mathcal{L}_{\mathrm{CE}}$ and $\mathcal{L}_{\mathrm{BCE}}$ often point in opposing directions. For example, a gradient step that reduces classification accuracy might enhance the detector's ability to identify adversarial examples. This phenomenon occurs because reducing classification accuracy involves adding perturbations to the input data, which in turn makes these perturbations more detectable by the adversarial detector. Consequently, the detector's task of distinguishing between clean and perturbed inputs becomes easier. This tug-of-war can impede the optimization process, preventing the simultaneous realization of both objectives. Lastly, the optimization landscapes of $\mathcal{L}_{\mathrm{CE}}$ and $\mathcal{L}_{\mathrm{BCE}}$ can dynamically shift as $\delta$ is updated.

To address this challenge we employ the selective and orthogonal approaches proposed by Bryniarski et al. (2021).

Selective Projected Gradient Descent (SPGD) (Bryniarski et al., 2021) optimizes only with respect to a constraint that has not been satisfied yet—while not optimizing against a constraint that has been optimized. The loss function used by SPGD is:

$$\mathcal{L}_{\mathrm{CE}}(f_{\theta}(x+\delta), y) \cdot \mathbb{1}[f_{\theta}(x) = y] + \mathcal{L}_{\mathrm{BCE}}(g_{\phi}(x+\delta), \mathtt{adv}) \cdot \mathbb{1}[g_{\phi}(x+\delta) = \mathtt{adv}]. \tag{9}$$

Rather than minimize a convex combination of the two loss functions, the idea is to optimize either the left-hand side of the equation, if the classifier's prediction is still $y$, meaning the optimization hasn't completed; or optimize the right-hand side of the equation, if the detector's prediction is still $\mathtt{adv}$.

This approach ensures that updates consistently enhance the loss on either the classifier or the adversarial detector, with the attacker attempting to "push" the classifier away from the correct class $y$, and also "push" the detector away from raising the adversarial "flag".

Orthogonal Projected Gradient Descent (OPGD) (Bryniarski et al., 2021) focuses on modifying the update direction to ensure it remains orthogonal to the constraints imposed by previously satisfied objectives. This orthogonal projection ensures that the update steers the input towards the adversarial objective without violating the constraints imposed by the classifier's decision boundary, allowing for the creation of adversarial examples that bypass the detector. Thus, the update rule is slightly different:

$$\mathcal{L}_{\text{update}}(x,y) = \begin{cases} \nabla \mathcal{L}_{\text{CE}}(f_\theta(x+\delta), y) - \text{proj}_{\nabla \mathcal{L}_{\text{BCE}}(g_\phi(x+\delta), \text{adv})} \nabla \mathcal{L}_{\text{CE}}(f_\theta(x+\delta), y) & \text{if} \quad f_\theta(x) = y \\ \nabla \mathcal{L}_{\text{BCE}}(g_\phi(x+\delta), \text{adv}) - \text{proj}_{\nabla \mathcal{L}_{\text{CE}}(f_\theta(x+\delta), y)} \nabla \mathcal{L}_{\text{BCE}}(g_\phi(x+\delta), \text{adv}) & \text{if} \quad g_\phi(x) = \text{adv}. \end{cases} \tag{10}$$

In essence we have set up a minimax game: the attacker wants to maximize loss with respect to the image, while the defender wants to minimize loss with respect to the detector's parameters.

Since optimization is customarily viewed as minimizing a loss function we will consider that the attacker uses minimization instead of maximization. A failed attack would thus be observed though a large loss value (which we will indeed observe in Section 5).

All aforementioned objectives are intractable and thus approximated using iterative gradient-based attacks, such as PGD. Algorithm 1 shows the pseudocode of our method for adversarially training an adversarial detector.

---

**Algorithm 1: RADAR**

---

**Input:** dataset $\mathcal{D}$, classifier $f_\theta$, detector, $g_\phi$, detector's loss function $\mathcal{L}_{\text{BCE}}$, classifier's loss function $\mathcal{L}_{\text{CE}}$, batch size $B$, step size $\alpha$, epsilon $\epsilon$, number of steps $I$, number of epochs $E$, learning rate $\eta$

**Output:** adversarially trained $g_\phi$

Set $f$ to eval mode

**for** $e = 1, ..., E$ **do**

    **for** $X_{ben} \subseteq \mathcal{D}, \ s.t. \ |X_{ben}| = B$ **do**

        Set $g$ to eval mode

        Compute adversarial examples $X_{\text{adv}}$:

$$X_{\text{adv}} \leftarrow \text{OPGD/SPGD}(f_\theta, g_\phi, X_{\text{ben}}, \epsilon, \alpha, I, \mathcal{L}_{\text{BCE}}, \mathcal{L}_{\text{CE}})$$

        Form a new batch:

$$X_{\text{mixed}} \leftarrow X_{\text{ben}} \oplus X_{\text{adv}}$$
$$Y_{\text{mixed}} \leftarrow \{\text{ben}\}_{i=1}^{|B|} \oplus \{\text{adv}\}_{i=1}^{|B|}$$

        Set $g$ to train mode

        Compute average loss gradient for $X_{\text{mixed}}$:

$$\nabla_{\text{mixed}} \leftarrow \frac{1}{2|B|} \sum_{j=1}^{2|B|} \nabla_\phi \mathcal{L}_{\text{BCE}}(x_{\text{mixed}}, y_{\text{mixed}}; \phi)$$

        Update parameters:

$$\phi \leftarrow \phi - \eta \nabla_{\text{mixed}}$$

    **end**

**end**

---

## 4 Experimental Framework

**Datasets and models**. We used the VGG-{11,13,16} and ResNet-{18, 34, 50} architectures both for classification and for adversarial detection. Specifically, we employed these architectures in dual roles: as classifiers to perform the primary task of image classification and as adversarial detectors to identify adversarial examples. By leveraging the same architectures for both tasks, we ensured consistency in our experimental setup and enabled a comprehensive evaluation of their robustness and effectiveness in the context of adversarial training. We modified the classification head of the detector, $C \in \mathbb{R}^{K \times e}$, where $K$ is the number of classes and $e$ is the embedding-space size, to $C \in \mathbb{R}^{2 \times e}$, for binary classification (benign/adversarial), and added a sigmoid transformation at the end.

Table 1: Performance (accuracy percentage) of original classifiers both on clean and adversarial, PGD-perturbed test sets.

| Dataset | VGG-11 | | VGG-13 | | VGG-16 | | ResNet-18 | | ResNet-34 | | ResNet-50 | |
|---------|--------|------|--------|------|--------|------|-----------|------|-----------|------|-----------|------|
| | Ben | Adv | Ben | Adv | Ben | Adv | Ben | Adv | Ben | Adv | Ben | Adv |
| CIFAR-10 | 92.40 | 0.35 | 94.21 | 0.19 | 94.00 | 5.95 | 92.60 | 0.14 | 93.03 | 0.21 | 93.43 | 0.26 |
| SVHN | 93.96 | 0.00 | 94.85 | 0.01 | 94.95 | 0.48 | 94.88 | 0.02 | 94.84 | 0.11 | 94.33 | 0.10 |
| ImageNet | 70.08 | 0.00 | 70.44 | 0.00 | 71.96 | 0.00 | 70.24 | 0.00 | 72.24 | 0.00 | 74.96 | 0.00 |

We utilized the CIFAR-10 (Krizhevsky, 2009), SVHN (Netzer et al., 2011) and a subset of ImageNet (Deng et al., 2009) dataset to evaluate our proposed approach. For ImageNet, we randomly selected 50 classes from the original dataset, which contains over 14 million images and 1,000 classes. This subset was used to maintain manageability and to focus our evaluation on a representative sample of the larger dataset, while still leveraging the diversity and complexity that ImageNet provides for benchmarking in image classification tasks. It is important to highlight that adversarial training for adversarial detectors incurs substantially higher computational costs compared to conventional adversarial training. As a result, this process is considerably more time-consuming when applied to large datasets. To ensure the feasibility of our experiments, we therefore employed a reduced number of classes from ImageNet.

We used the definition of attack success rate presented by Bryniarski et al. (2021) as part of their evaluation methodology. Attack efficacy is measured through Attack Success Rate at N, SR@N: proportion of deliberate attacks achieving their objectives subject to the condition that the defense's false-positive rate is configured to N%. The underlying motivation is that in real-life scenarios we must strike a delicate balance between security and precision. A 5% false positive is acceptable, while extreme cases might even use 50%. Our results proved excellent so we set N to a low 5%, i.e., SR@5.

Throughout the training, the allowed perturbation was constrained by an $\ell_\infty$ norm, denoted as $\| \cdot \|_\infty$, with a maximum magnitude set to $\epsilon = \frac{16}{255}$. We employed an adversarial attack strategy, specifically utilizing 100 iterations of PGD with a step size parameter $\alpha$ set to 0.03. This approach was employed to generate adversarial instances and assess the resilience of the classifier under scrutiny. We then evaluated the classifiers on the attacked test set—the results are delineated in Table 1.

Afterwards, we split the training data into training (70%) and validation sets (30%). We trained 3 VGG-based and 3 ResNet-based adversarial detectors for 20 epochs, using the Adam optimizer (with default $\beta_1$ and $\beta_2$ values), with a learning rate of $1e - 4$, CosineAnnealing learning-rate scheduler with $T_{\max} = 10$, and a batch size of 32. We tested the adversarial detectors on the test set that was comprised of one half clean images and one half attacked images: They all performed almost perfectly.

Following the initial clean training phase, we implemented our proposed RADAR approach, which involves adversarial fine-tuning on the adversarial detectors. This process entailed attacking the detectors to an adaptive adversarial attack, specifically the OPGD attack. The adversarial fine-tuning was conducted over the course of 20 epochs, utilizing the same optimizer and the ReduceLROnPlateau learning rate scheduler, with the patience parameter set to 3, and a batch size of 32. Subsequently, the performance was evaluated on a test set consisting of an equal distribution of clean images and images subjected to OPGD attacks.

## 5 Results

Initially, we conducted an assessment of classifier performance following the generation of adversarial perturbations employing the PGD method.

The outcomes of this evaluation, presented in Table 1, highlight classifier performance on both clean and perturbed test sets. The results reveal significant vulnerability to adversarial manipulations. For instance, on the CIFAR-10 dataset, the VGG-11 classifier drops from 92.40% accuracy on clean data to 0.35% on adversarial examples. Other classifiers on CIFAR-10 show similar trends, with clean accuracy values between

Table 2: **Without** `RADAR`: Performance of detectors on several classifiers and datasets. In this and subsequent tables, boldface marks best performance. Note: for ROC-AUC, higher is better.

| Detector | $\text{AUC}_{\text{Avg.}}$ | CIFAR-10 | | | SVHN | | | ImageNet | | |
|---|---|---|---|---|---|---|---|---|---|---|
| | | $\text{AUC}_{\text{PGD}}$ | $\text{AUC}_{\text{OPGD}}$ | $\text{AUC}_{\text{SPGD}}$ | $\text{AUC}_{\text{PGD}}$ | $\text{AUC}_{\text{OPGD}}$ | $\text{AUC}_{\text{SPGD}}$ | $\text{AUC}_{\text{PGD}}$ | $\text{AUC}_{\text{OPGD}}$ | $\text{AUC}_{\text{SPGD}}$ |
| VGG-11 | 0.33 | 1.00 | 0.00 | 0.00 | 1.00 | 0.00 | 0.00 | 1.00 | 0.00 | 0.00 |
| VGG-13 | 0.33 | 1.00 | 0.00 | 0.00 | 1.00 | 0.00 | 0.00 | 1.00 | 0.00 | 0.00 |
| VGG-16 | **0.46** | 1.00 | 0.61 | 0.59 | 1.00 | 0.00 | 0.00 | 1.00 | 0.00 | 0.00 |
| ResNet-18 | 0.33 | 1.00 | 0.00 | 0.00 | 1.00 | 0.00 | 0.00 | 1.00 | 0.00 | 0.00 |
| ResNet-34 | 0.33 | 1.00 | 0.00 | 0.00 | 1.00 | 0.00 | 0.00 | 1.00 | 0.00 | 0.00 |
| ResNet-50 | 0.37 | 1.00 | 0.15 | 0.14 | 1.00 | 0.03 | 0.05 | 1.00 | 0.00 | 0.00 |
| **Avg.** | 0.36 | 1.00 | 0.13 | 0.12 | 1.00 | 0.00 | 0.00 | 1.00 | 0.00 | 0.00 |

Table 3: **Without** `RADAR`: Performance of detectors on several classifiers and datasets. VGG-16 is the most robust detector in terms of SR@5. Note: for SR@5, lower is better.

| Detector | $\text{SR@5}_{\text{Avg.}}$ | CIFAR-10 | | SVHN | | ImageNet | |
|---|---|---|---|---|---|---|---|
| | | $\text{SR@5}_{\text{OPGD}}$ | $\text{SR@5}_{\text{SPGD}}$ | $\text{SR@5}_{\text{OPGD}}$ | $\text{SR@5}_{\text{SPGD}}$ | $\text{SR@5}_{\text{OPGD}}$ | $\text{SR@5}_{\text{SPGD}}$ |
| VGG-11 | 0.98 | 0.97 | 0.97 | 0.98 | 0.99 | 0.99 | 1.00 |
| VGG-13 | 0.99 | 0.97 | 0.99 | 0.99 | 0.99 | 1.00 | 1.00 |
| VGG-16 | **0.78** | 0.37 | 0.40 | 0.99 | 0.99 | 0.97 | 0.99 |
| ResNet-18 | 0.98 | 0.96 | 0.97 | 0.99 | 0.99 | 0.99 | 1.00 |
| ResNet-34 | 0.98 | 0.96 | 0.97 | 0.99 | 0.99 | 0.99 | 1.00 |
| ResNet-50 | 0.93 | 0.81 | 0.83 | 0.96 | 0.94 | 0.97 | 0.99 |
| **Avg.** | 0.90 | 0.84 | 0.85 | 0.98 | 0.98 | 0.98 | 0.99 |

92.40% and 94.21%, and adversarial accuracy values below 6%. A similar pattern is observed on the SVHN dataset, where the VGG-11 classifier's accuracy falls from 93.96% on clean data to 0.00% on adversarial attacks. Other SVHN classifiers also exhibit significant performance drops, with adversarial accuracy values near 0%. On the ImageNet dataset, all classifiers reached 0% accuracy when attacked. These findings illustrate the stark vulnerability of standard classifiers to adversarial perturbations, underscoring the need for more robust defense mechanisms.

After standard adversarial training involving the use of PGD-generated adversarial images, Table 2 shows the performance outcomes of the adversarial detectors, as assessed through the ROC-AUC metric, before the deployment of `RADAR`. The table summarizes the average ROC-AUC ($\text{AUC}_{\text{Avg.}}$) and individual ROC-AUC scores for PGD ($\text{AUC}_{\text{PGD}}$), OPGD ($\text{AUC}_{\text{OPGD}}$), and SPGD ($\text{AUC}_{\text{SPGD}}$) attacks. For the CIFAR-10 dataset, VGG-16 detector achieved the highest average ROC-AUC score of 0.46, which is slightly worse than tossing a coin. All detectors performed perfectly against PGD attacks, but their performance dropped significantly against OPGD and SPGD, with most detectors showing a score close to 0.00. The SVHN and ImageNet datasets displayed a similar trend. All models achieved perfect detection against PGD attacks, but their effectiveness plummeted for OPGD and SPGD attacks, with only ResNet-50 showing minimal detection capability.

Table 3 presents results regarding the adversarial detectors' performance, based on the SR@5 metric, prior to using `RADAR`. Notably, the VGG-16 detector exhibits the best performance with a significantly lower $\text{SR@5}_{\text{Avg.}}$ value of 0.78. It performs particularly well against OPGD and SPGD attacks on CIFAR-10, achieving SR@5 values of 0.37 and 0.40, respectively. This indicates a higher robustness compared to other detectors. In contrast, detectors such as VGG-11, VGG-13, ResNet-18, and ResNet-34 show higher $\text{SR@5}_{\text{Avg.}}$ values around 0.97 for both datasets. ResNet-50 performs better than most with an average SR@5 of 0.88 on CIFAR-10 but still falls short of VGG-16. On the ImageNet dataset, the detectors also failed to withstand the attacks, achieving SR@5 values of 0.98 and 0.99 for OPGD and SPGD, respectively.

We then deployed `RADAR`. The outcomes on the efficacy of adversarial detectors, measured by ROC-AUC and SR@5 metrics after integrating `RADAR`, are presented in Table 5 and Table 6. Table 4 shows the accuracy

percentages of all the classifiers on clean and adversarially perturbed test sets across CIFAR-10, SVHN, and ImageNet datasets, after applying `RADAR`. Notably, we constructed an equal distribution of clean and adversarially perturbed test sets, effectively doubling the original test-set size to ensure a balanced evaluation. Accuracy was computed as follows: If the detector identified the input as adversarial, the classifier's prediction was disregarded; otherwise, the classifier's prediction was considered. The observed accuracy drop in Table 4, particularly noticeable in smaller models, can be attributed to their limited number of parameters, making adaptation to both adaptive and standard adversarial attacks more challenging. Note that the benign accuracy for all classifiers across CIFAR-10, SVHN, and ImageNet remained unchanged.

For all datasets, `RADAR` maintains high accuracy both on PGD and OPGD samples. VGG-11 on CIFAR-10 retains an accuracy of 92.40% across all scenarios, while ResNet-50 on SVHN shows only a slight reduction from 94.33% to 94.32% on OPGD samples. In contrast, the ImageNet dataset reveals more significant accuracy drops under adversarial conditions, particularly for ResNet-18, which falls from 70.24% on benign samples to 61.05% on PGD samples.

Table 4: **With `RADAR`**: The accuracy percentage of the original classifiers is assessed both on clean and adversarially perturbed test sets, using both PGD and OPGD, subsequent to the application of `RADAR`. In instances where the adversarial detector indicates an adversarial sample (`adv`), the classifier's prediction is disregarded.

| Dataset | VGG-11 | | | VGG-13 | | | VGG-16 | | | ResNet-18 | | | ResNet-34 | | | ResNet-50 | | |
|---|---|---|---|---|---|---|---|---|---|---|---|---|---|---|---|---|---|---|
| | Ben | PGD | OPGD | Ben | PGD | OPGD | Ben | PGD | OPGD | Ben | PGD | OPGD | Ben | PGD | OPGD | Ben | PGD | OPGD |
| CIFAR-10 | 92.40 | 92.40 | 92.37 | 94.21 | 94.17 | 94.08 | 94.00 | 93.96 | 93.91 | 92.60 | 92.41 | 91.84 | 93.03 | 93.03 | 93.03 | 93.43 | 93.43 | 93.14 |
| SVHN | 93.96 | 93.96 | 93.95 | 94.85 | 94.85 | 94.85 | 94.95 | 94.95 | 94.94 | 94.88 | 94.88 | 94.86 | 94.84 | 94.83 | 94.82 | 94.33 | 94.33 | 94.32 |
| ImageNet | 70.08 | 68.89 | 68.73 | 70.44 | 69.87 | 69.83 | 71.96 | 71.96 | 71.96 | 70.24 | 61.05 | 68.63 | 72.24 | 68.18 | 72.18 | 74.96 | 74.63 | 74.96 |

Following `RADAR` integration, we observed significant improvements in adversarial detection performance across various classifiers and datasets, as shown in Table 5. Notably, on CIFAR-10, models such as VGG-13, VGG-16, ResNet-34, and ResNet-50 achieved high average ROC-AUC scores of 0.99, with ResNet-18 slightly lower at 0.98. For SVHN, all models achieved ROC-AUC scores of 1.00, except for ResNet-18 and ResNet-50, which scored 0.99 under specific attack conditions. Similarly, for the ImageNet dataset, models exhibited robust performance, with most achieving ROC-AUC scores of 1.00. Specifically, ResNet-50 scored consistently high at 1.00 or 0.99 across different attack types. These results underscore `RADAR`'s ability to fortify adversarial detectors against various attack methods.

The SR@5 metric further confirms the robustness of `RADAR`-enhanced detectors, as detailed in Table 6. Models like VGG-13, VGG-16, ResNet-34, and ResNet-50 achieved perfect SR@5 scores of 0.00 on all datasets, indicating successful detection of adversarial attacks. VGG-11 and ResNet-18 demonstrated slightly lower performance on CIFAR-10. `RADAR` significantly improves the robustness of adversarial detectors, as evidenced by high ROC-AUC scores and low SR@5 values across different models and datasets, including the challenging ImageNet dataset. This comprehensive performance across diverse datasets reinforces the efficacy of our proposed approach in enhancing adversarial detector resilience.

Figure 2 show the generalization performance of `RADAR`-trained detectors on classifiers they were not trained on. This table shows the ROC-AUC values of `RADAR`-trained detectors, when evaluated on the different classifier models. Each cell in the table corresponds to the ROC-AUC value achieved by the detector-classifier pair. For example, the top-left cell in the top-right table indicates the performance of a VGG-11 detector model trained on a VGG-11 classifier model using the ImageNet dataset, and subsequently evaluated on attacks utilizing a ResNet-50 classifier model. The results indicate consistently high generalization across different classifiers, with most detector-classifier pairs achieving ROC-AUC values of 0.99 or higher. This demonstrates the effectiveness of `RADAR`-trained detectors in maintaining high adversarial detection performance across diverse datasets and classifier architectures. These findings underscore the resilience of adversarial detectors trained with `RADAR`, showcasing their ability to maintain robust detection capabilities even when confronted with unseen classifiers. The high ROC-AUC values indicate that our approach effectively fortifies the detectors themselves, rather than relying solely on robust classifier training, thereby enhancing the overall security and reliability of the adversarial detection system.

Table 5: **With `RADAR`:** Performance of employing `RADAR` with OPGD, on several classifiers and datasets.

| Detector | $\text{AUC}_{\textbf{Avg.}}$ | CIFAR-10 | | | SVHN | | | ImageNet | | |
|---|---|---|---|---|---|---|---|---|---|---|
| | | $\text{AUC}_{\text{PGD}}$ | $\text{AUC}_{\text{OPGD}}$ | $\text{AUC}_{\text{SPGD}}$ | $\text{AUC}_{\text{PGD}}$ | $\text{AUC}_{\text{OPGD}}$ | $\text{AUC}_{\text{SPGD}}$ | $\text{AUC}_{\text{PGD}}$ | $\text{AUC}_{\text{OPGD}}$ | $\text{AUC}_{\text{SPGD}}$ |
| VGG-11 | 0.98 | 0.99 | 0.95 | 0.94 | 1.00 | 1.00 | 1.00 | 1.00 | 1.00 | 1.00 |
| VGG-13 | **0.99** | 0.99 | 0.99 | 0.99 | 1.00 | 1.00 | 1.00 | 1.00 | 0.99 | 0.99 |
| VGG-16 | **0.99** | 1.00 | 0.99 | 0.99 | 1.00 | 1.00 | 1.00 | 1.00 | 0.99 | 0.99 |
| ResNet-18 | 0.98 | 0.99 | 0.96 | 0.96 | 1.00 | 0.99 | 1.00 | 0.95 | 1.00 | 1.00 |
| ResNet-34 | **0.99** | 0.99 | 0.99 | 1.00 | 1.00 | 1.00 | 1.00 | 0.98 | 1.00 | 1.00 |
| ResNet-50 | **0.99** | 0.99 | 0.99 | 0.99 | 1.00 | 1.00 | 0.99 | 0.99 | 1.00 | 1.00 |
| **Avg.** | 0.99 | 0.99 | 0.98 | 0.98 | 1.00 | 0.99 | 0.99 | 0.98 | 0.99 | 0.99 |

Table 6: **With `RADAR`:** Performance of detectors on several classifiers and datasets.

| Detector | $\text{SR@5}_{\textbf{Avg.}}$ | CIFAR-10 | | SVHN | | ImageNet | |
|---|---|---|---|---|---|---|---|
| | | $\text{SR@5}_{\text{OPGD}}$ | $\text{SR@5}_{\text{SPGD}}$ | $\text{SR@5}_{\text{OPGD}}$ | $\text{SR@5}_{\text{SPGD}}$ | $\text{SR@5}_{\text{OPGD}}$ | $\text{SR@5}_{\text{SPGD}}$ |
| VGG-11 | 0.02 | 0.04 | 0.06 | 0.00 | 0.00 | 0.00 | 0.00 |
| VGG-13 | **0.00** | 0.00 | 0.00 | 0.00 | 0.00 | 0.00 | 0.00 |
| VGG-16 | **0.00** | 0.00 | 0.00 | 0.00 | 0.00 | 0.00 | 0.00 |
| ResNet-18 | 0.02 | 0.05 | 0.05 | 0.00 | 0.00 | 0.00 | 0.00 |
| ResNet-34 | **0.00** | 0.00 | 0.00 | 0.00 | 0.00 | 0.00 | 0.00 |
| ResNet-50 | **0.00** | 0.00 | 0.00 | 0.00 | 0.00 | 0.00 | 0.00 |
| **Avg.** | 0.00 | 0.01 | 0.02 | 0.00 | 0.00 | 0.00 | 0.00 |

A notable enhancement is observed across all detectors with respect to ROC-AUC and SR@5. Moreover, our findings suggest that adversarial training did not optimize solely for adaptability to specific adversarial techniques, but also demonstrated efficacy against conventional PGD attacks.

**Impact of adversarial training on optimization dynamics.** Before the incorporation of adversarial training, the optimization process exhibited a trend of rapid convergence towards zero loss within a few iterations, as can be seen in the top rows of Figure 3, Figure 4 and Figure 5. Once we integrated adversarial training into the detector, we observed a distinct shift in the optimization behavior. Upon deploying `RADAR`, the optimization process displayed a tendency to plateau after a small number of iterations, with loss values typically higher by orders of magnitude, as compared to those observed prior to deployment. This plateau phase persisted across diverse experimental settings and datasets, indicating a fundamental change in the optimization landscape, induced by `RADAR`. This behavior can be seen as a robustness-enhancing effect, because the detector appears to resist rapid convergence towards trivial solutions, thereby enhancing its generalization capabilities. This showcases the efficacy of our method in bolstering the defenses of detection systems against adversarial threats—without sacrificing clean accuracy.

**Robustness analysis with various epsilon values.** The results of our experiments, depicted in Figure 6 and Figure 7, provide a comprehensive evaluation of the robustness of the adversarial detectors against OPGD and SPGD attacks with different $\epsilon$ values.

For the CIFAR-10, SVHN, and ImageNet datasets, increasing $\epsilon$ values generally results in increasing ROC-AUC scores and decreasing SR@5 rates, using both OPGD and SPGD, indicating improved detection capabilities and decreased success of adversarial attacks.

Specifically, VGG-11 shows a significant drop in AUC and a corresponding rise in SR@5 as $\epsilon$ decreases, reflecting its susceptibility to lower magnitude perturbations. VGG-13 and VGG-16 exhibit similar patterns, though VGG-16 demonstrates slightly better robustness at lower $\epsilon$ values. This trend is consistent across both OPGD and SPGD attack evaluations.

The ResNet models, particularly ResNet50, demonstrate superior resilience performance. Across various $\epsilon$ values, ResNet50 consistently exhibits higher ROC-AUC scores and lower SR@5 rates compared to other models, signifying its robust ability to detect subtle adversarial examples. Notably, our method maintains

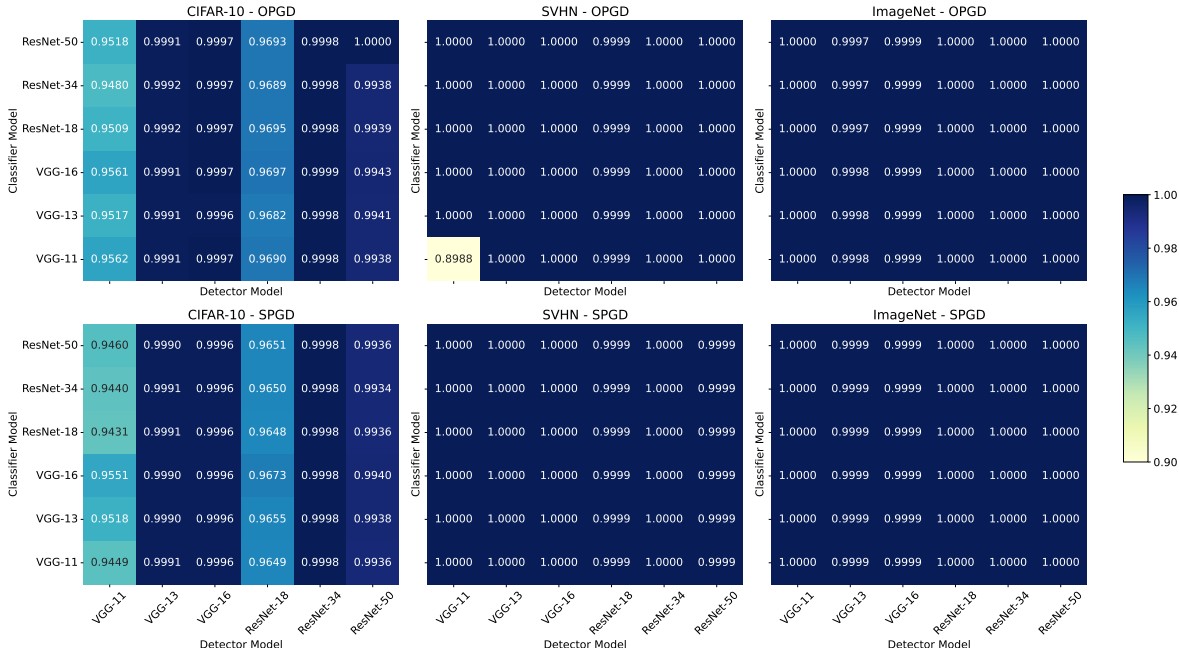

Figure 2: Generalization performance of adversarially trained detectors, trained on CIFAR-10, SVHN and ImageNet. Each adversarial detector was trained using each corresponding classifier, e.g. ResNet-50 adversarial detector was trained using ResNet-50 image classifier. This table shows the generalization of each detector to other classifiers, which it didn't train with. A value represents the ROC-AUC of the respective detector/classifier pair, for OPGD (top row) and SPGD (bottom row) with $\epsilon = \frac{16}{255}$.

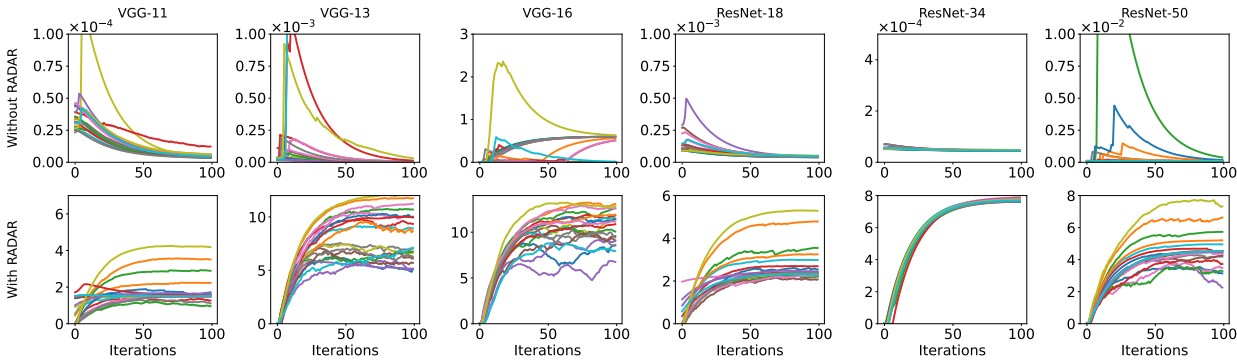

Figure 3: CIFAR-10. Binary cross-entropy loss metrics, from the point of view of an attacker, are herein presented in the context of crafting an adversarial instance from the test set. These plots illustrate the progression of loss over 20 different images of orthogonal projected gradient descent (OPGD), with the main goal being to minimize the loss. Top: Prior to adversarial training, the loss converges to zero after a small number of iterations. Bottom: After adversarial training, the incurred losses are significantly higher by orders of magnitude (note the difference in scales), compared to those observed in their standard counterparts. This shows that the detector is now resilient, i.e., far harder to fool.

consistently low SR@5 values across all datasets, underscoring its resilience. Specifically, experiments on the SVHN dataset consistently achieve an SR@5 of 0%. For CIFAR-10 and ImageNet, SR@5 ranges between 0% to 17% with OPGD, and between 0% to 15% with SPGD, affirming the efficacy of our approach in enhancing adversarial-detection capability. These results highlight the advantage of applying adversarial training directly to adversarial detectors rather than focusing solely on classifiers themselves.

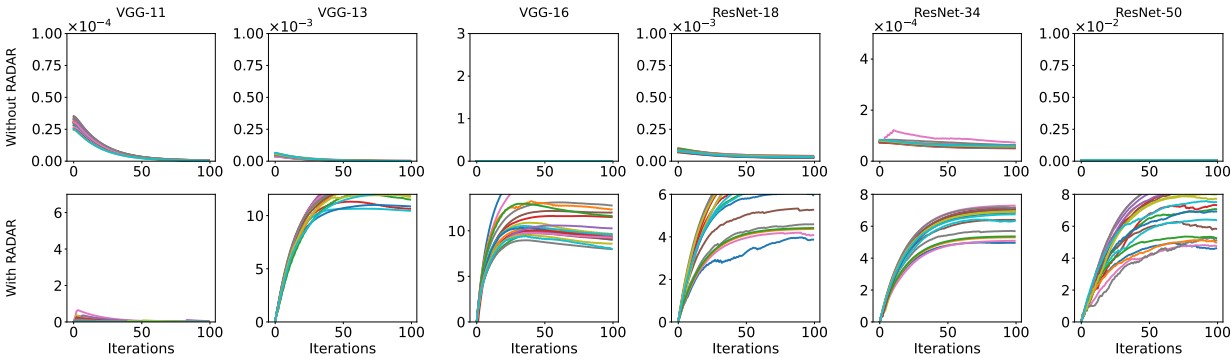

Figure 4: SVHN. Binary cross-entropy loss metrics, from the point of view of an attacker, are herein presented in the context of crafting an adversarial instance from the test set.

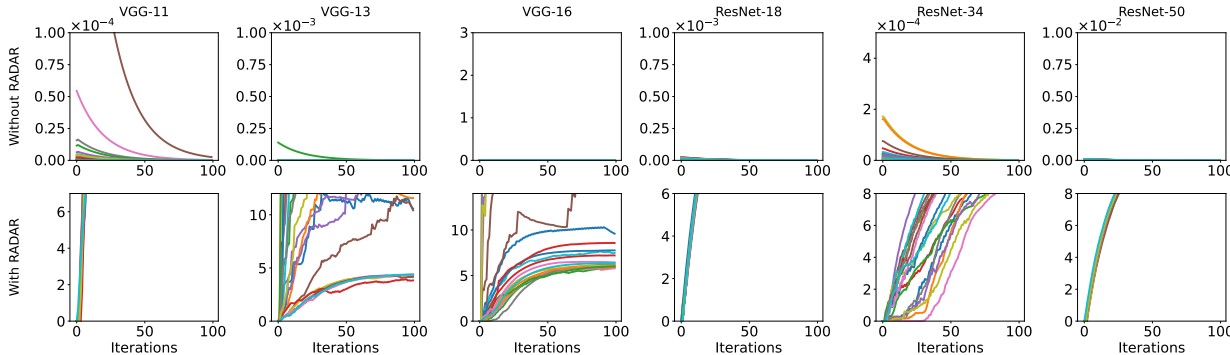

Figure 5: ImageNet. Binary cross-entropy loss metrics, from the point of view of an attacker, are herein presented in the context of crafting an adversarial instance from the test set using OPGD.

**Comparison to other detectors against adaptive attacks.** The table presents the robust accuracy ($\text{Acc}_{\text{robust}}$) of various defenses against OPGD and SPGD adaptive attacks at two perturbation levels, $\epsilon = 0.01$ and $\epsilon = 8/255$, and at two false positive rates, 5% (FP@5) and 50% (FP@50). The defenses evaluated are ContraNet (Yang et al., 2022b), Trapdoor (Shan et al., 2020), DLA (Sperl et al., 2020), and SID (Tian et al., 2021). `RADAR` consistently achieves the highest robust accuracy across both attack types and perturbation levels, maintaining nearly 100% accuracy in all scenarios. ContraNet also shows high performance, particularly at lower perturbation levels, but its accuracy decreases at higher perturbation levels. Trapdoor, DLA, and SID exhibit varying degrees of robustness, with significant decreases in accuracy at higher perturbation levels. For instance, Trapdoor's accuracy drops to almost 0% under both attack types at $\epsilon = 8/255$.

## 5.1 Ablation Studies

In this section, we conduct comprehensive ablation studies to evaluate the impact of critical hyperparameters in our proposed loss function on the performance of our adversarial detectors on the validation set. Specifically, we focus on four key parameters: number of steps, step size ($\alpha$), batch size, and learning rate. Each of these parameters plays a significant role in the training process and potentially influences the robustness and effectiveness of our adversarial detectors. To evaluate the effectiveness and trade-offs associated with these parameters, we conducted a series of experiments using the ResNet-50 architecture on the CIFAR-10 dataset. The results are delineated in Figure 8.

The results reveal that the number of steps significantly affects the model's performance. The detector's performance reaches its peak at 100 steps. This suggests that more iterations in the adversarial training process lead to better optimization and enhanced robustness of the adversarial detector. The choice of $\alpha$

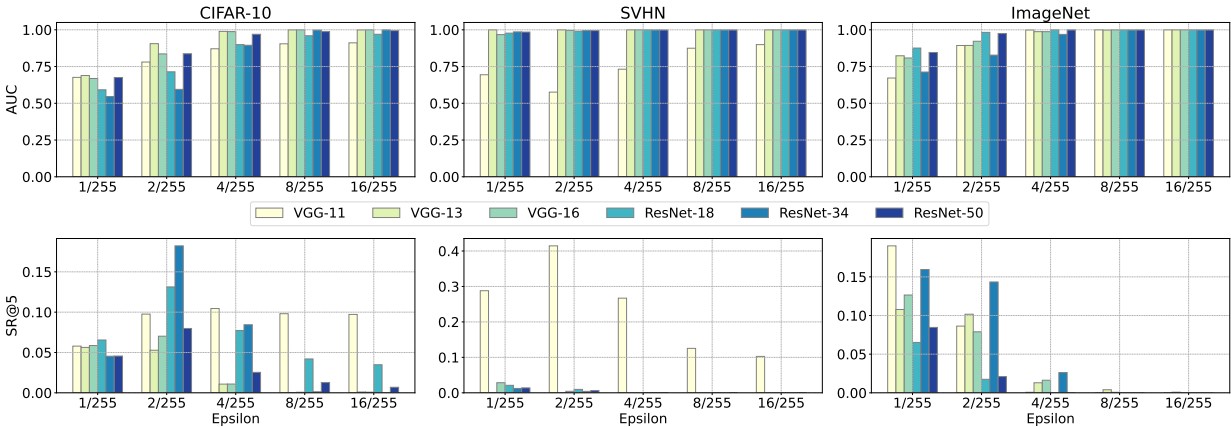

Figure 6: AUC and SR@5 scores across different epsilon values for CIFAR-10, SVHN, and ImageNet datasets using OPGD. The performance of the adversarial detectors is illustrated, highlighting how AUC and SR@5 varies across different perturbation magnitudes.

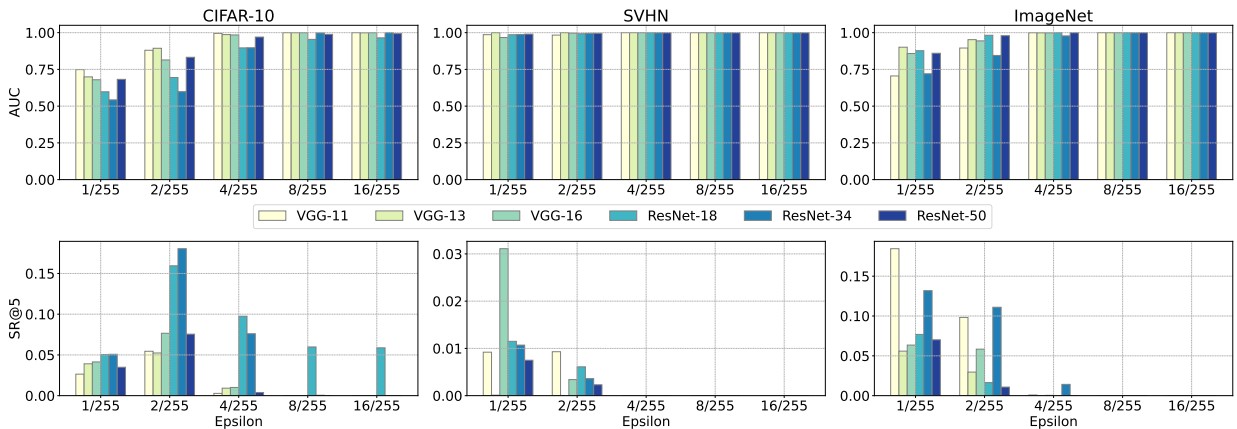

Figure 7: AUC and SR@5 scores across different epsilon values for CIFAR-10, SVHN, and ImageNet datasets using SPGD.

also plays a critical role. Our findings show that a smaller $\alpha$ (0.05) yields the best performance, while increasing $\alpha$ increases effectiveness. This indicates that bigger perturbations during training are more effective in strengthening the detector's resilience to adversarial attacks. But we want to detect attacks that are unrecognizable, thus we chose to use $\alpha = 0.03$.

Regarding batch size, the performance remains relatively stable across smaller batch sizes but shows a notable decline at the largest batch size tested (256).

Learning rate is another crucial factor. Initially, a learning rate of $1 \times 10^{-1}$ and $1 \times 10^{-2}$ performs worse, but as the learning rate decreases, there is an improvement in performance. However, a learning rate that is too small ($1 \times 10^{-5}$) results in under-performance, highlighting the need for a balanced learning rate to ensure effective training.

Based on this ablation study, we selected the hyperparameters that yielded the best performance: a batch size of 32, a learning rate of $1 \times 10^{-2}$, 100 steps, and an $\alpha$ value of 0.03. These settings were found to optimize the robustness of our adversarial detector, providing a strong defense against adversarial attacks.

Table 7: Robust accuracy of different defenses under OPGD and SPGD attacks. The table presents the robust accuracy ($\text{Acc}_{\text{robust}}$) of various defenses when subjected to the adaptive attacks, OPGD and SPGD. The accuracy is evaluated at two perturbation levels, $\epsilon = 0.01$ and $\epsilon = 8/255$, and is reported for two false positive rates (FP), 5% (FP@5) and 50% (FP@50).

| Attack | Defense | $\epsilon = 0.01$ | | $\epsilon = 8/255$ | |
|--------|---------|-----------------------------|------------------------------|-----------------------------|------------------------------|
| | | $\text{Acc}_{\text{robust}}FP@5$ | $\text{Acc}_{\text{robust}}FP@50$ | $\text{Acc}_{\text{robust}}FP@5$ | $\text{Acc}_{\text{robust}}FP@50$ |
| OPGD | **RADAR** | **99.2%** | **99.3%** | **99.8%** | **99.9%** |
| | ContraNet (Yang et al., 2022b) | 93.7% | 99.2% | 89.8% | 94.7% |
| | Trapdoor (Shan et al., 2020) | 0.0% | 7.0% | 0.0% | 8.0% |
| | DLA (Sperl et al., 2020) | 62.6% | 83.7% | 0.0% | 28.2% |
| | SID (Tian et al., 2021) | 6.9% | 23.4% | 0.0% | 1.6% |
| SPGD | **RADAR** | **99.2%** | **99.1%** | **99.8%** | **99.9%** |
| | ContraNet (Yang et al., 2022b) | 93.7% | 99.3% | 89.7% | 95.1% |
| | Trapdoor (Shan et al., 2020) | 0.2% | 49.5% | 0.4% | 37.2% |
| | DLA (Sperl et al., 2020) | 17.0% | 55.9% | 0.0% | 13.5% |
| | SID (Tian et al., 2021) | 8.9% | 50.9% | 0.0% | 11.4% |

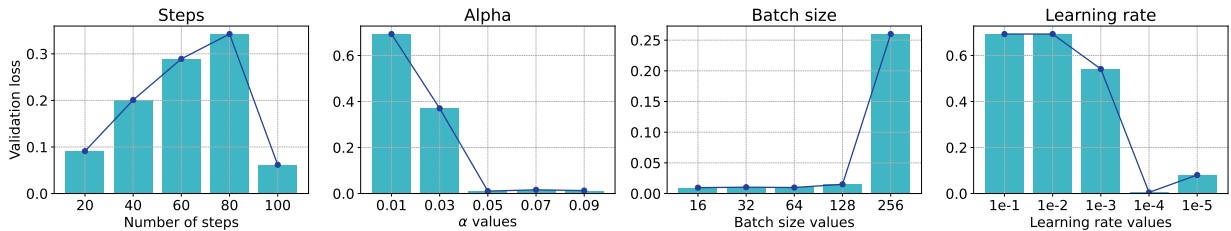

Figure 8: Ablation studies using VGG-11 with different number of steps, $\alpha$, batch size, and learnig rate values, from left to right.

As illustrated in Figure 9, we conducted an in-depth analysis of the impact of excluding standard training and relying solely on adversarial training across three distinct datasets: CIFAR-10, SVHN, and ImageNet.

For CIFAR-10, the results demonstrate that standard training significantly boosts performance—particularly for ResNet models—at lower epsilon values (1/255 and 2/255). Accuracy for ResNet-18, for example, shows a clear advantage with standard training, highlighting its role in enhancing model robustness at smaller perturbations. However, as the epsilon values increase, the gap between models trained with and without standard training narrows, suggesting that adversarial training alone is sufficient to maintain robustness at higher perturbations.

In the case of SVHN, the benefits of standard training are evident in ResNet-based models but less pronounced compared to CIFAR-10. Accuracy trends show that while standard training does provide a marginal improvement—particularly noticeable for the ResNet architectures—models trained exclusively with adversarial training still achieve good performance. This suggests that for simpler datasets, e.g., SVHN, the reliance on standard training might be reduced without significant loss in robustness, especially at mid-to-high epsilon values.

For ImageNet, results show a relatively modest impact with standard training compared to the other datasets. Performance improvement when incorporating standard training is not as pronounced, which may be attributed to the fact that all models in this evaluation were pretrained on ImageNet. This pretraining likely provided the models with a strong baseline robustness, diminishing the relative benefit of standard training in this specific case. As with the other datasets, when the epsilon values increase, the performance gap between models with and without standard training narrows further, suggesting that adversarial training alone is effective at higher perturbation strengths.

Overall, the analysis across these datasets indicates that while standard training contributes positively to model robustness, particularly at lower epsilon values, adversarial training alone can achieve comparable performance as the perturbation strength increases. The decision to incorporate standard training may therefore depend on the specific dataset and the computational resources available.

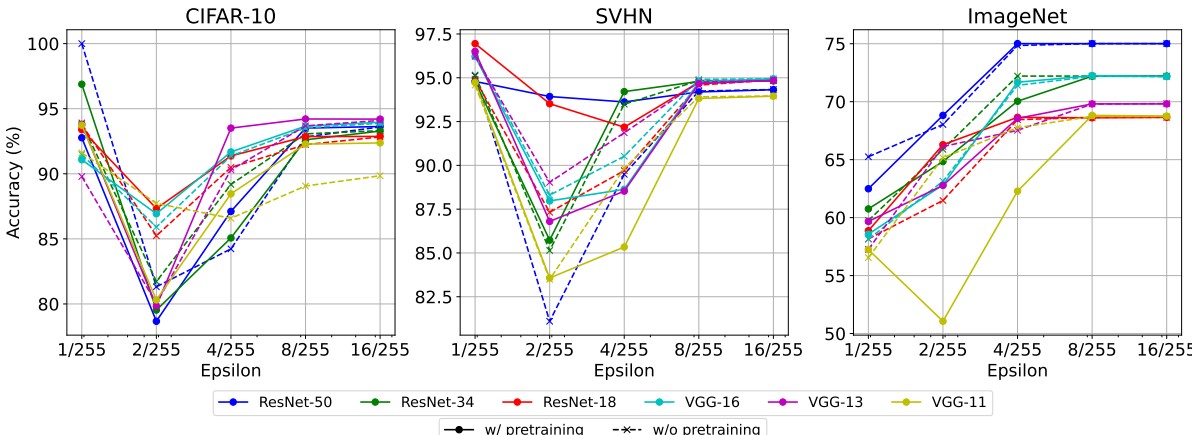

Figure 9: Comparison of classification accuracy of models utilizing adversarial detectors trained with standard pretraining followed by adversarial training (solid lines) versus detectors trained exclusively with adversarial training (dashed lines) across CIFAR-10, SVHN, and ImageNet datasets. The plot demonstrates the impact of increasing $\epsilon$ values on the accuracy of the classification models when using the respective adversarial detectors.

# 6 Conclusions

We presented a paradigmatic shift in the approach to adversarial training of deep neural networks, transitioning from fortifying classifiers to fortifying networks dedicated to adversarial detection. Our findings illuminate the prospective capacity to endow deep neural networks with resilience against adversarial attacks. Rigorous empirical inquiries substantiate the efficacy of our developed adversarial training methodology.

**There is no trade-off between clean and adversarial classification because the classifier is not modified.**

Our results bring forth, perhaps, a sense of optimism regarding the attainability of adversarially robust deep learning detectors. Of note is the significant robustness exhibited by our networks across the datasets examined, manifested in increased accuracy against a diverse array of potent $\infty$-bound adversaries.

Our study not only contributes valuable insights into enhancing the resilience of deep neural networks against adversarial attacks but also underscores the importance of continued exploration in this domain. Therefore, we encourage researchers to persist in their endeavors to advance the frontier of adversarially robust deep learning detectors.

**Acknowledgments**

Anonymized.

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
