# OpenReview forum: "Fortify the Guardian, Not the Treasure: Resilient Adversarial Detectors"
_TMLR — Rejected by TMLR_

### Review · Reviewer_UcBv · 2024-06-05

**Summary Of Contributions:**

The paper introduces RADAR, a method to improve adversarial detectors' robustness against adaptive attacks while maintaining classifier performance. The approach uses adversarial training to reinforce the ability to detect attacks without compromising clean accuracy. The method integrates adversarial exercises into the dataset, allowing the detector to learn and adapt to potential attack scenarios. Experimental evaluations show that the algorithm significantly improves detector accuracy in identifying adaptive attacks.

**Audience:**

Yes

**Claims And Evidence:**

Yes

**Requested Changes:**

1. Page 6:
> While gradient descent allows for simultaneous maximization of the cross-entropy loss (LCE) and the binary cross-entropy loss (LBCE), they might contradict each other, adopting different optimization strategies.

Why LCE and LBCE might contradict each other? It is not clear to me.

2. Page 6, Section 4
> Datasets and models. We used the VGG-{11,13,16} and ResNet-{18, 34, 50} architectures both for classification and for adversarial detection.

Do you use VGG and ResNet both as classifiers and detectors? This point is not clear.

3. Section 5: As I mentioned, the explanation and description of the experimental findings are quite limited. The readers typically don't read the figures and tables and discover something interesting by themselves.

**Strengths And Weaknesses:**

Stengths:
+ The proposed method is specifically designed to improve the robustness of adversarial detectors against adaptive attacks.
+ By incorporating adversarial training into the approach, the method reinforces the detector's ability to recognize and respond to attacks.

Weaknesses
- The method seems like a simple application of OPGD on adversarial detectors, which means the novelty might be limited.
- The contributions listed in Section 1, IMHO, seem not like big ones. For instance, robustifying the adversarial detector is more like a natural idea in adversarial training. Also, RADAR, is like an application of OPGD/SPGD on the minimax game.
- The experiment is not with an ablation study, which may hinder its effectiveness explanation
- Furthermore, the description of experimental results is limited. As a reader, I would expect more discussion on the experimental findings.

---

### Review · Reviewer_D54q · 2024-06-05

**Summary Of Contributions:**

This paper proposed an approach to resolve the adversarial attack in image classification. Instead of making the classifier be robust, the authors trained another robust adversarial image detector, then, the if the detector finds the image is an adversarial image, flag it and do not make the prediction from the original classifier; otherwise, the classifier will make the prediction. With this approach, the trade-off between accuracy and robustness is better compared to standard adversarial training.

**Audience:**

Yes

**Claims And Evidence:**

Yes

**Requested Changes:**

1. Comparing to other adversarial detector works, even though those methods are not co-trained with classifier.

2. Experiments with larger dataset, e.g. imagenet1k.

3. The classification accuracy with RADAR rather than just showing the detector results.

Minor:

1. Please define K in eq. 1.
2. Layout of section 5, the tables can be all put at the top instead of interleaving the texts between tables.

**Strengths And Weaknesses:**

Strengths:

1. The proposed method trades off between computations/parameters and performace by separating the detection and classification into two stages. With, good detector, the proposed method can achieve high accuracy in classification.

2. The paper is easy to follow.

Weaknesses:
1. There is no comparison to any existed works on adversarial image detection.
e.g., New Adversarial Image Detection Based on Sentiment Analysis, IEEE TNNLS. or the one in the related works.

2. Only small-scale datasets and models are evaluated.

3. At high-level, comparing to the models trained with adversarial training, the proposed method needs much more computations (depending on the complexity of detector).

---

### Review · Reviewer_fsSt · 2024-07-23

**Summary Of Contributions:**

This paper proposes an adversarial training method for the attack detector to enhance its robustness against adaptive adversarial attacks. Instead of training the classifier to be robust, the paper improves the robustness of the attack detector, which it claims to avoid the accuracy-robustness tradeoff. Adaptive attacks generated with OPGD/SPGD are used in the training of the detector. Experimental results claim the proposed method can still perfectly detect adversarial examples under the threat of adaptive attack.

**Audience:**

Yes

**Broader Impact Concerns:**

No broader impact concerns.

**Claims And Evidence:**

No

**Requested Changes:**

As mentioned in the weakness part, the following changes are suggested to improve paper quality and justify the contribution
1. Please revise Sec. 3.2, especially the contents on page 6, to clearify the discussion about the optimization objective is for the attack optimization
2. Add ablation study to verify the effectiveness of the initial clean trianing phase
3. Explain how accuracy is computed under the experiment setting and the source of accuracy drop in Tab. 4
4. Report classifier accuracy against adaptive attack with a smaller strength
5. Report the generalization of the detector on transfer adversarial attack generated on other surrogate models

**Strengths And Weaknesses:**

## Strength
1. This paper is among the few that investigate the adversarial training of the attack detector rather then the classifier. The impact of adversarial training on the detector is less studied. Results in this line can be inspiring to future work.
2. The paper considers SOTA adaptive attack generation methods in the design of the adversarial training algorithm
3. Experiments are done on a diverse dataset settings including ImageNet.

## Weakness
1. Sec. 3.2 is confusing, especially for the paragraphs after Equ. (8). It appears that this section want to explain the optimization process of the adaptive attack, but int he text the discussion is about updating model weight. A full revision of this section is needed.
2. The proposed method uses the same architecture of the classifier as the detector, effectively doubles the inference cost. It would be interesting to see the effectiveness of smaller detector architecture.
3.  The detector follows a 2-stage training, yet the impact of the initial clean training phase is not disucssed and justified
4. The "accuracy" reported in Tab. 4 is confusing. How exactly the accuracy is computed is not clear. Furthermore, given the almost 100% AUC reported in Tab. 5, it is unclear why significant accuracy drop is observed on ImageNet models in Tab. 4.
5. From Fig. 6, the detector is not performing well against an attack with a smaller epsilon. As the classifier may be fooled by an attack of small strength, the worst accuracy may not be achieved with the largest strength attack as reported in previous tables.
6. Only adaptive attacks used in the adversarial training process are used for evaluation, which does not support the generalizability of the proposed method. Transfer attack and other attack generation methods are needed to rule out the potential of overfitting and gradient masking.

---

### Decision · Action_Editor_CaPY · 2024-09-23

**Recommendation:** Reject

**Comment:**

The paper proposes RADAR for improving adversarial detector robustness. In summary, I believe it lacks the evidence to back up its claims due to limited evaluation, and there are unclear explanations, making it unlikely to engage TMLR's audience.

**Audience:**

The reviewers questioned the broader relevance of the work, with UcBv calling the experimental findings "quite limited," and D54q highlighting concerns about computational overhead which is a concern if this is to be adopted. Overall I feel the paper’s contributions to the field are limited, making it unlikely to engage TMLR’s audience.

**Claims And Evidence:**

I believe it fails to meet TMLR’s acceptance criteria of having clear, convincing evidence. As Reviewer fsSt noted, sections like 3.2 are "confusing" and "need a full revision." Reviewer UcBv pointed out that the method appears to be "a simple application of OPGD/SPGD," limiting any claims of novelty. Additionally, Reviewer fsSt pointed out overfitting concerns, stating that "the detector is not performing well against an attack with a smaller epsilon," suggesting the method may not generalize beyond this specific attack.

Further, the paper's evaluation is restricted to small datasets like CIFAR-10 and a subset of ImageNet, leading Reviewer D54q to state that this "loses the purpose to demonstrate its scalability" for larger, real-world scenarios. The paper also lacks evaluations against a broader range of attack types, which fsSt commented on, stating, "only adaptive attacks used in the adversarial training process are used for evaluation."

The authors did include additional experiments to show generalization to attacks from other surrogate models. While this was an effort to address the concern, it was too limited in scope I believe to show that it's applicable to a wider range of attacks.

**Resubmission Of Major Revision:**

The authors may consider submitting a major revision at a later time.